# Application of Single-Cell and Spatial Omics in Musculoskeletal Disorder Research

**DOI:** 10.3390/ijms24032271

**Published:** 2023-01-23

**Authors:** Site Feng, Jiahao Li, Jingjing Tian, Sheng Lu, Yu Zhao

**Affiliations:** 1School of Medicine, Tsinghua University, Beijing 100084, China; 2Department of Orthopaedic Surgery, Peking Union Medical College Hospital, Peking Union Medical College and Chinese Academy of Medical Science, Beijing 100730, China; 3Medical Science Research Center, Peking Union Medical College Hospital, Peking Union Medical College and Chinese Academy of Medical Sciences, Beijing 100730, China; 4The Key Laboratory of Digital Orthopaedics of Yunnan Provincial, Department of Orthopedic Surgery, The First People’s Hospital of Yunnan Province, The Affiliated Hospital of Kunming University of Science and Technology, Kunming 650032, China

**Keywords:** single cell, spatial omics, musculoskeletal disorders, fracture, dysplasia, heterotopic ossification, osteoporosis, osteoarthritis, intervertebral disc, tendinopathy

## Abstract

Musculoskeletal disorders, including fractures, scoliosis, heterotopic ossification, osteoporosis, osteoarthritis, disc degeneration, and muscular injury, etc., can occur at any stage of human life. Understanding the occurrence and development mechanism of musculoskeletal disorders, as well as the changes in tissues and cells during therapy, might help us find targeted treatment methods. Single-cell techniques provide excellent tools for studying alterations at the cellular level of disorders. However, the application of these techniques in research on musculoskeletal disorders is still limited. This review summarizes the current single-cell and spatial omics used in musculoskeletal disorders. Cell isolation, experimental methods, and feasible experimental designs for single-cell studies of musculoskeletal system diseases have been reviewed based on tissue characteristics. Then, the paper summarizes the latest findings of single-cell studies in musculoskeletal disorders from three aspects: bone and ossification, joint, and muscle and tendon disorders. Recent discoveries about the cell populations involved in these diseases are highlighted. Furthermore, the therapeutic responses of musculoskeletal disorders, especially single-cell changes after the treatments of implants, stem cell therapies, and drugs are described. Finally, the application potential and future development directions of single-cell and spatial omics in research on musculoskeletal diseases are discussed.

## 1. Introduction

The musculoskeletal system, consisting of the bones, muscles, cartilage, tendons, ligaments, and other connective tissues, supports the body, allows motion, and protects the vital organs [1]. Although we already know that bones develop to form the skeleton through a series of synchronized events [2], the cell populations involved in the development of skeletal, muscle, and connective tissue disorders are incompletely understood. Obviously, abnormal musculoskeletal conditions occur throughout the life course, from childhood to older age. They can be either short-term (fractures, sprains, and strains), which cause pain and functional restrictions, or long-term (osteoarthritis (OA) and chronic primary lower back pain). Many cell populations are implicated in the recovery progression of these disorders. The function, expression, and molecular signaling pathways of different cell populations in the recovery process can provide targets for treatment [2,3,4,5,6].

In addition, implants such as artificial bones and artificial joints are routinely used in the surgical treatment of skeletal diseases. Less understood, but equally important, are the interactions between implants and living cells and tissues [7]. Understanding the characteristics of the peri-implant cell population can reveal the causes of implant-related complications, including aseptic inflammation, loosening, and other problems. It provides a theoretical basis for researchers to select implant materials and design structures. However, due to the complicated anatomical nature and such a complex meshwork of different cell types, progress on these issues has been hampered by limitations inherited from traditional histological analysis methods for a long time [8]. Otherwise, stem cells and drugs for the treatment of musculoskeletal system diseases are also a major focus of current research [9,10,11]. Understanding the local cell changes after the intervention of these treatments will help us to discover the treatment mechanism of the disease and thus inspire the development of new drugs.

Since the first description of scRNA-seq in 2009 [12], single-cell technology has contributed to significant advances both in the discovery of new cell types and groundbreaking insights into a variety of disorders. Spatial omics techniques were later developed to decode spatial context of tissues and cells. In recent years, knowledge about single-cell isolation and identification for the musculoskeletal system has continuously accumulated [13,14]. By offering crucial insights into the processes that maintain and regenerate the musculoskeletal system during homeostasis and repair, single-cell and spatial omics technologies allow researchers to observe accurate delineation of cellular activity and destiny throughout musculoskeletal system growth and repair.

This review will summarize recent discoveries generated by single-cell and spatial omics techniques in the musculoskeletal system which reshape our understanding of musculoskeletal system conditions. This article will focus on new discoveries in the field of disease and therapy from the design of single-cell and spatial omics studies of the musculoskeletal system, involving cell characteristics and interactions between cells and implants, thus informing on the current state of science and potential gaps in knowledge.

## 2. Experimental Design

Most single-cell characterizations of the musculoskeletal system are currently limited to the transcriptome. While single-cell multi-omics (specifically cellular indexing of transcriptomes and epitopes by sequencing, CITE-seq) [15,16], cytometry by time of flight (CyTOF) [17,18], and spatial omics [19,20] have only been applied to a few studies, single-cell RNA-seq (scRNA-seq) is a common strategy applied to characterize musculoskeletal-associated transcriptomes, allowing high-resolution views of cellular heterogeneity within specific and well-characterized cell populations. Over the past decade, spatial transcriptomics techniques have been developed to fill the gap between classic techniques that maintain spatial information, such as immunofluorescence or *in situ* hybridization (ISH), and novel methodology, such as scRNA-seq, which can simultaneously query the whole transcriptome [21]. The invention of these new approaches, as well as other spatial-omics techniques, has facilitated novel discoveries in diverse fields, including the skeletal system, which could be concertedly performed with scRNA-seq. In this section, we will discuss the experimental designs which take advantage of single-cell and spatial transcriptomics when profiling the musculoskeletal system.

### 2.1. Single-Cell and Spatial Transcriptomics Techniques and Downstream Analyses

The currently available scRNA-seq technologies can be divided into two main categories according to different strategies of cell isolation, droplet-based [22,23,24] and plate-based [25,26,27,28]. Droplet-based methods support massively parallel sequencing of ~10,000 cells and are thus advantageous in rare cell type recognition, while plate-based protocols lead to a lower number of cells per experiment and subsequently increased costs per cell. Conversely, plate-based protocols have higher sensitivity in capturing gene expression features, referred to as sequencing depth, while droplet-based methods are less capable of distinguishing subtle differences among cells [29]. The characteristics of these technologies together indicate an effective complementary strategy [29]. The complex tissues in the musculoskeletal system are first characterized by high-breadth droplet-based technologies to find novel populations and related markers, which may then be employed in high-depth, plate-based techniques for deep sequencing [29]. Most of the current studies in the musculoskeletal system have taken the droplet-based approach, while only a few studies have used plate-based strategies [30].

Spatial transcriptomics techniques are categorized into five strategies, including highly multiplexed single-molecule FISH (smFISH), single molecular in situ sequencing (ISS), region of interest (ROI), next-generation sequencing (NGS) with spatial barcodes, and methods not requiring prior spatial locations [31]. Among these techniques, smFISH has the highest detection efficiency, whereas NGS barcoding techniques have a relatively low efficiency. Techniques such as ROI, NGS, and untargeted ISS do not target a specific set of genes, thus covering a wide range within the transcriptome. Techniques with lower detection efficiencies, especially NGS techniques, are likely to cover larger tissue areas. SmFISH- and ISS-based techniques perform a single-cell and even single-molecule spatial resolution which goes beyond the capability of most NGS and ROI techniques [31]. The most commercial and easily accessible platforms include LCM [32] and Nanostring GeoMX [33] (ROI methods), 10X Visium [34] (NGS with spatial barcodes), and Cartana ISS (an ISS method) [31]. Currently, only NGS was used in studies on musculoskeletal diseases [19,20]. Since many of the above methods are complementary to each other, we expect other methods to be applied, such as MERFISH [35], STARmap [36], and GeoMX [33].

Downstream analyses start by characterizing or localizing distinct cell subsets. Compared to traditional methods of cell type identification based on a limited repertoire of markers, such as flow cytometry, single-cell sequencing techniques assemble a large amount of molecular expression information to determine the cell type [29]. Similarly, spatial omics techniques detect a larger number of molecules simultaneously than previous spatial profiling methods, enabling the discovery of co-expressed molecules in a pathological region and thus identifying the cell types and pathological pathways that are enriched in that region [31]. Differentially expressed genes among cell subsets and spatially variable genes are identified using canonical statistical tests [29,31].

Trajectory analysis is often used to describe pseudo-time inference of unsynchronized single-cell studies by constructing a calculated differentiation trajectory according to the differential gene expression among cell states [37]. Among advanced tools for trajectory construction, Monocle [38] has a widespread usage, while similar methods such as TSCAN [39], Slingshot [40], SLICE [41], and PAGA [42] are not commonly adopted in the research of diseases. The primary drawback of Monocle and most other trajectory analyses is that they cannot perform differential analysis among different conditions, such as between diseased and healthy samples. Recently published methods, Condiment [43] and PhenoPath [44], imperfectly solved the problem, and more powerful tools are yet to be designed. In practical clinical scenarios, it is useful to consider the spatial structure of the tissue in order to decode the disease’s progression. Differences in gene expression between spatial points are usually correlated with differences in time points as the disease expands along a fixed direction of development or region of infection. In spatial transcriptomics, Tower et al. [19] raised the concept of “spatial time” to measure the gene expression patterns along the spatial direction that calvarium stem cells migrate through in the process of differentiation.

Pathway enrichment condenses the extensive gene list of interest, usually produced by differential expression analysis into a pathway list of interest [45]. In research on musculoskeletal diseases, using tools such as gene set enrichment analysis (GSEA) [45,46] and gene set variation analysis (GSVA) [47], gene lists are aligned with various pathway databases, preferably (GO) and KEGG. Numerous thousands of standardized terminologies for biological processes (BP), molecular functions (MF) [45], and cellular components (CC) are provided by GO in a hierarchically ordered framework [45]. The KEGG database comprises a variety of pathways, including canonical signaling pathways and disease-associated gene sets rather than typical pathways [45].

Cell–cell communication (CCC) describes the interactions of cell excretes and cell surface molecules, which is the basic process in the biological reshaping of tissue and can be inferred from either single-cell data or spatial transcriptomics data [48]. Tools, including CellChat [49] and CellPhoneDB [50], have already been adopted by several musculoskeletal studies. CCC networks predicted by scRNA-seq data are likely to contain false-positive links due to a lack of spatial information, whereas CCC is limited to cells in close proximity [48]. By combining scRNA-seq with cutting-edge ST technologies that preserve spatial information, this restriction can be overcome, although frequently at the expense of cellular resolution, coverage, or sequencing depth. These techniques, such as SpaOTsc [48], have not yet been used in musculoskeletal disorders.

### 2.2. Biomarkers Applied in the Single-Cell Characterization of the Musculoskeletal System

The skeletal lineage includes osteoblasts, osteocytes, and chondrocytes, which are involved mainly in forming bone and cartilage, whereas hematopoietic lineage-derived osteoclasts (OC) contribute to bone resorption. The differentiation of skeletal lineage cells requires many steps from stem cells. Mesenchymal stromal cells (or mesenchymal stem cells, MSCs) are multipotent stem cells that give rise to all cell types in the skeletal lineage, and skeletal stem cells (SSCs) give rise to osteoblasts [2]. In previous single-cell studies of the human musculoskeletal system, researchers used various transcript markers to identify cell identities, as concluded in Table 1. Recently, a number of single-cell atlases profiling the musculoskeletal system of either mice or humans have recognized heterogeneous cell subtypes within stem cells and skeletal lineage cells using scRNA-seq [13,51]. The advancement of single-cell techniques will enable more studies to characterize healthy or diseased human samples, thus promoting the accuracy of cell type identification and facilitating the recognition of potentially disease-driving subsets.

### 2.3. Advanced Protocol for Isolating Qualified Single Cells

In both droplet-based and plate-based strategies, sophisticated protocols for isolating qualified single cells are essential to achieve high cell viability, high cell yield, and low content of cell aggregates or doublets [60]. Protocols for musculoskeletal tissue dissociation often use mechanical dissection, enzymatic degradation, and combinatorial techniques. Unlike non-adherent cells such as peripheral blood mononuclear cells, musculoskeletal tissues in their diseased condition are tense and thick, making it difficult to effectively remove the extracellular matrix, reduce cell damage, and generate single-cell suspensions. Cell isolation from bones, cartilage, nucleus pulposus (NP), and tendons was formerly performed using different enzymatic digestion methods, but quality control was often overlooked [60,61]. With the usage of an inappropriate endosteal cell recovery protocol, Ayturk et al. concluded from their study that scRNA-seq is not as sensitive as bulk RNA-seq for identifying changes in osteoblast gene expression [62]. About 10,000 genes were aligned from bulk RNA-seq across millions of cells per animal, while 75 osteoblasts per animal were collected from scRNA-seq samples, with each cell expressing about 1500 genes [62]. This further indicates the importance of constructing a suitable cell separation strategy.

Gao et al. reported an improved protocol in human sample dissociation adopting the S-D solution (STEMxyme1and Dispase II) originally designed by Baryawno et al. [63], who reported the application of the solution in single-cell RNA sequencing (scRNA-seq) analysis of mice bone and bone marrow. In an improved S-D solution, optimized concentrations of STEMxyme1, dispase II, pronase, collagenase type I and collagenase type II in solutions were designed for NP, ossifying posterior longitudinal ligaments, OA articular cartilage, respectively [60]. We found that dissociation solutions introduced in other human single-cell studies contain one or several of the enzymes mentioned in the S-D solution, usually together with DNase I. Further optimization of the enzymatic digestion protocol is expected in the future. Increasing the sample size will also amplify changes in the distribution of cell types following genetic, pharmacologic, or environmental perturbations [62].

### 2.4. Strategies for Hypothesis Generation and Validation in Single-Cell Omics under Different Sample Collection Situations

The resource of diseased human samples is limited, with them mostly coming from clinical surgeries. Apart from affected tissues as experimental groups, unaffected tissues such as the intervertebral disc (IVD) or cartilage are also accessible as control groups. For instance, healthy IVDs with complete nucleus pulposus (NP), annulus fibrosus (AF), and the cartilaginous endplate (CEP) structures were acquired from patients with brain death, whereas NP alone can be acquired from spine fractures and spinal disease [55].

Single-cell transcriptomic atlases profiling healthy tissues alone naturally lend themselves to hypothesis generation. According to the prior knowledge of gene functions, clusters with specific gene markers are likely to perform corresponding functions. Single-cell research on healthy samples mainly aims to predict potential disease-driving clusters. For instance, the current research on osteoporosis is limited to healthy primary human femoral head tissue cells (FHTCs) [54], in which osteoclast and metabolism-related cell clusters are likely to drive this disease. The usage of pseudo-time trajectory analysis and cell–cell interaction analysis adds more predictions about the relationship between cells within a single type of sample.

When samples are accessible from both disease and control tissues, researchers observe the differences in cell composition and recognize differentially expressed genes (DEG) from which pathway enrichment should be performed using tools such as GSEA and GSVA in databases such as GO and KEGG. Most sample studies used this method, especially in IVD degeneration and vertebral fracture patients undergoing fusion surgery, as these patient samples are easily acquired from surgeries.

As for developmental disease, human samples remain difficult to obtain; thus, easy-to-manipulate transgenic mice play an essential role in research on the lineage specification process, primarily the (Cre)–*loxP* system [2]. Other methods of constructing rat models include material implantation, chemical damage, and so on. By comparing differentially expressed genes and different cell properties between the affected and control mice, and by enriching potentially functioning pathways, single-cell techniques further investigate previously determined molecules or pathways involved in disorders, such as the Hedgehog, Notch, WNT, BMP, and FGF signaling pathways [2].

Bones and muscles, together with their accessory structures, form complex architecture; thus, spatial omics techniques are particularly useful in decoding their biological processes. Complicated musculoskeletal structures, such as the calvarium and joints, have already been characterized by this advanced method to compare differential spatial gene expression between affected tissues and healthy ones [19,20].

## 3. Single-Cell and Spatial Omics Were Applied to the Characterization of Various Musculoskeletal Diseases

Musculoskeletal disorders in different organs undergo various pathologies, with different cell lineages involved. In this section, we summarize the diseases into osteogenic, chondrogenic, or joint-related and muscle or tendon-related ones. The causes or phenotypes of the diseases include acute injuries, developmental malfunctions, degenerations, and heterotopic ossifications (Figure 1, Table 2, Table 3 and Table 4). Single-cell and spatial omics techniques add knowledge to various diseases, whereas further analyses of these diseases, or even a broader range of disorders, are expected to be applied.

### 3.1. Bone-Related Disorders

#### 3.1.1. Bone Injury

The body’s capacity to heal bones via regeneration, returning them to their completely functioning and pre-injury form, demonstrates the importance of bones to mammalian physiology [2]. Bone fracture healing is a complex process that consecutively undergoes the inflammatory phase, two healing phases of soft and hard callus formation, and a remodeling phase [83]. During the process, the two main cellular components of bones, bone-forming osteoblasts, and bone-resorbing osteoclasts, exhibit a controlled balance of activities [2]. Research gaps still need to be filled in identifying regeneration-driving clusters and molecular mechanisms in different stages and locations, and single-cell analysis still needs to be fully applied. 

Regarding the ribs, single-cell RNA sequencing of soft callus tissue generated after rib injury in mice with defective Hedgehog signaling revealed a decrease in Cxcl12-expressing cells, which indicates a failure to attract Cxcl12-expressing skeletal stem and progenitor cells (SSPCs) during the regenerative response [64]. Regarding the long bones in the leg, scRNA-seq profiling of non-hematopoietic stromal cells was performed on mice at post-fracture day 14, compared to age-matched, unfractured controls. The cell proportions of chondrocytes, fibroblasts, and Fabp5^+^ Mmp9^+^ septoclasts (SCs) [84,85,86] increased in fractured bones, while that of diaphyseal MSCs (dpMSCs) decreased. Trajectory analysis further suggests the conversion of dpMSCs into mpMSCs, which leads to the expansion of osteoblast lineage cells, fibroblasts, chondrocytes, and SCs [87].

In cranial bone injuries, Xu et al. [65]. found that nerve growth factor (NGF), a previously identified factor inducing skeletal reinnervation [88,89], is expressed in cranial bone injuries. NGF signals via p75 (*Ngfr*) [90] in resident mesenchymal osteogenic precursors to affect their migration into the damaged tissue. The researchers collected the injured frontal bones of mice with p75 conditionally knockout in mesenchymal cells. Single-cell transcriptomics identified repair- and inflammatory-response-related clusters that are affected by p75. A subcluster of mesenchymal cells that expressed high amounts of Itgb1 was enriched for specific GO pathways, including cell adhesion, modulation of actin cytoskeleton structure, and cell migration. Genes involved in receptor-mediated extracellular matrix interaction and cytokine-receptor-mediated signaling, both of which decreased in p75-deficient people, were significantly enriched in the clusters. Another crucial mesenchymal subcluster expressed a significant quantity of inflammatory regulators, such as Il1a, Il10, and Tnf, whose expression decreased in cells produced from mice lacking p75. Pathway analysis of the whole mesenchymal cell populations revealed that osteogenesis-related signaling pathways, including Notch, bone morphogenetic protein, transforming growth factor, fibroblast growth factor, and Hippo signaling, were suppressed in p75-deficient cells [65].

Research on bone injuries provides us with a deeper understanding of the cell populations and cytokines associated with different bone injuries. This provides a theoretical basis for us to solve challenging bone defects by means of tissue engineering scaffolds.

#### 3.1.2. Skeletal Dysplasia

The developmental malformations of the musculoskeletal system contain congenital anomalies and post-natal diseases, including the abnormal development of the flat bone or long bone, joint dysplasia, and spinal deformity [66]. In normal congenital bone development, two types of bone ossification, intramembranous and endochondral, occur at different locations. Although these processes start with mesenchymal stem cells (MSC), their transformations into bones are distinct. Intramembranous ossification transforms mesenchymal tissue directly into the bone and forms the flat bones of the skull, clavicle, and the majority of the cranial bones. Endochondral ossification in long bones starts with the transformation of mesenchymal tissue into hypertrophic cartilage, which is thereafter replaced by bone [2,91,92]. These processes are followed by forming the axial skeleton and the long bones in post-natal development [2,91,93]. Although plenty of single-cell analyses have characterized normal skeletal developmental processes in mice [94], only a limited number of studies used single-cell or spatial omics to propose the potential pathology in skeletal dysplasia.

Congenital skeletal dysplasia is caused by genetic mutations that impair the pattern, structure, and development of bones [66]. This dysplasia appears as one or more phenotypes that impact the form and size of specific skeletal locations, such as short, stubby fingers, duplications of fingers or toes, clubfeet, missing bones, fragile bones, or curved spines [66]. Current single-cell studies to the characterization of mice hind limbs and calvarium have focused on molecules, pathways, and even the temporal and spatial nature that cause malformations.

From E15.5 hind limb tissues, Wang et al. [67] predicted the crucial function of IHH signaling in the transformation from cartilage to bone. Hypertrophic chondrocytes (HCs) were identified by the expression of early or late HC-specific markers, including Col10a1, Ihh, Mmp13, Vegfa, etc., and crucial transcriptional regulators such as Zbtb20 and Runx2. They used pseudo-temporal cell trajectory analysis to determine transcriptional patterns in HCs and Col1a1^+^ Runx2^+^ osteoblasts, with HCs positioned at the start point of the pseudo-time trajectory and Col1a1^+^ Runx2^+^ osteoblasts at the end of the trajectory. Analysis of Hedgehog gene expression through the trajectory showed that Ptch1 was expressed momentarily and was independent of Gli1-3 expression [95]. These findings, taken together, point to a distinctive pattern of hedgehog-related transcriptome states at the single-cell level during the simulated cartilage-to-bone transition. Further histological images on the mice model verified that deleting *Ptch1* in HCs affected the formation of primary spongiosa and HC-derived osteogenic cells, which led to the bony bulges in adult mutant animals.

Through spatial transcriptomic data from murine calvarium, Tower et al. [19] used an approach similar to the scRNAseq computational technique of inferring pseudo-time, referred to herein as “spatial time” to compare changes in gene expression between tropomyosin receptor kinase A (TrkA) mutant and control mice. This methodology measures and normalizes the distance between each spatial place in a suture and a manually chosen central anchor point in the midline to the suture width. The midline suture, in particular, showed a low expression of numerous previously known factors, including Gli1, Twist1, and noggin, which were expressed and/or implicated in maintaining suture patency. The BMP/TGF signaling pathway components were dysregulated in the calvariae of TrkA mutants, according to the pathway analysis. BMP activation was specifically enriched within the osteogenic front (OF), with little activation within the suture midline, according to the modular scoring used to evaluate the expression of BMP activators and inhibitors; BMP inhibitory scores revealed the opposite spatial localization. Similar outcomes were shown for TGF signaling. These findings, to some extent, indicated that some extracellular factors might have affected the molecular pathways of cells, and several secreted factors were later identified, including FSTL1 [19].

Idiopathic scoliosis is a prevalent developmental deformity of the spine [96]. The etiology of idiopathic scoliosis is not yet understood but is likely to relate to skeletal and paravertebral muscle abnormalities. Potential causes of adolescent idiopathic scoliosis (AIS) include genetic, environmental, endocrinological, metabolic, biochemical, neurological, and asymmetric growth variables [52,97], suggesting that both congenital and post-natal factors might contribute to this abnormality.

Yang et al. [52] acquired scRNA-seq data from healthy and diseased samples to demonstrate the features of MSC, chondrocyte progenitor cells (CPC), and osteoblasts (OC). In the three identified MSC subtypes, *IGFBP5*-expressing MSC (MSC-IGFBP5) was identified as an AIS-specific MSC subtype. The cell number of osteoblasts was significantly decreased in AIS subjects, possibly due to the failure of MSC-IGFBP5’s differentiation into osteoblasts, the decreased potential of cell proliferation, and increased cell death. Regarding CPC, *PCNA*-expressing CPC was the specific subtype recognized only in AIS patients. The cell counts of *BIRC3*-expressing OC in AIS were less than that in the controls. Pseudo-time analysis suggested distinct patterns of osteoclast differentiation. In AIS, monocytes differentiated into CRISP3-expressing OC, which is different from the trajectory of the control subjects [52].

Due to the difficulty in obtaining clinical tissues of developmental malformations, most of the current single-cell studies are based on animal models, and their conclusions are usually further verified by histological or immunofluorescence imaging. In addition, existing studies mainly focus on the malformation of the skeletal system. Although the normal developmental processes of cartilage and joints have been characterized by several studies in mice [98,99], there are currently no single-cell studies on abnormal cartilage and joint development being reported. Future research can apply the single-cell methods to the characterization of other developmental diseases in other organs.

#### 3.1.3. Heterotopic Ossification

Heterotopic ossification (HO) refers to ectopic endochondral bone growth and accumulation inside or around connective tissues, muscles, joints, blood vessels, and other locations, which is a frequent and potentially disabling acquired condition [100,101,102]. Tendon ossification, ossification of the ligamentum flavum (OLF), and ossification of the posterior longitudinal ligament (OPLL) are common HO of tendon and ligament (HOTL) diseases that disable patients. The HO of the above diseases occurs in the spinal cord or nerves, causing pain, paralysis, or even death due to compression [103,104]. Metabolomics studies have identified changes in certain cellular pathways in HO [105,106]. HO is an abnormal change in local tissues, and single-cell techniques are helpful for researchers to find the changes in cell expression in local tissues.

By implanting rhBMP2–Matrigel mixtures subcutaneously into mice and injecting a pre-immune anti-body (BMP2/IgG) or an activin A neutralizing antibody (BMP2/nActA.Ab), and ordering cells in a continuous two-dimensional pseudo-time trajectory, Mundy et al. revealed that activin A (encoded by Inhba), a member of the TGF-β superfamily, stimulated HO development in muscles [68]. Fewer cells in the early stages of the pseudo-time trajectory and more in the terminal (Acan- and Col2a1-expressing) segment were seen in the BMP2/IgG samples than in the BMP2/nActA samples. Furthermore, progenitors expressing Inhba in the BMP2/IgG and BMP2/nActA.Ab samples filled the latter part of the trajectory, while Matrigel-only samples were restricted to the early part. These findings suggest that activin A may promote heterotopic ossification and that activin-A-neutralizing antibody therapy prevents the formation of intramuscular HO [68].

A fraction of tendon stem/progenitor cells (TSPCs) expressing lubricin (PRG4) is believed to contribute to tissue repair [69]. The processes underlying ectopic ossification and their connections with TSPC, however, are yet unknown. Tachibana et al. explored the features of Prg4^+^ cells and discovered that a unique Prg4^+^ TSPC cluster expresses R-spondin 2 (RSPO2), a WNT activator. Pseudo-time trajectory characterized the Rspo2^+^ cluster as an undifferentiated one, and predicted that the cluster might differentiate into downstream tenogenic and chondro/osteogenic clusters [107].

At present, the process of ectopic ossification is still unclear to researchers. Different sites of ectopic ossification have similar histological changes. Is there a similar or identical mechanism? Perhaps with the help of single-cell technology, we may be able to answer that question in the future.

#### 3.1.4. Osteoporosis

Osteoporosis is a prevalent disorder characterized by a decreased bone mineral density and an increased risk of osteoporotic fractures [108,109]. Importantly, bone homeostasis disturbance plays a critical role in the etiology of osteoporosis. As a highly metabolically active tissue, bones involve a continual cycle of bone formation by osteoblasts and bone resorption by osteoclasts [108]. Notably, the regulation of bone mineral density is highly heritable, and disorders in gene expression result in osteoporosis and osteoporotic fracture; nevertheless, decoding the underlying genomic and molecular mechanisms of osteoporosis in vivo in humans remains problematic [108,109].

Samples of osteoporosis subjects can hardly be collected from the surgeries in the current studies. Wang et al. [54] performed single-cell RNA analysis on primary human femoral head tissue cells (FHTCs) from healthy subjects. Predictions were acquired from the pseudo-time trajectory of osteoclast formation and the enrichment of possible transcript biomarkers and signaling pathways that probably contribute to osteoporosis. Expression of potential critical genes, such as *OLMF4*, *RPL39*, *H3F3B*, and *SAT1*, altered dramatically throughout the progressive trajectory of OC formation. The transcriptional factors, including *HMGB2*, *HMGB1*, *MEF2C*, *ID1*, *ID3*, *LITAF*, and *CREM*, associated with immune cell proliferation and differentiation, were increasingly dysregulated as the trajectory differentiation process progressed. The zinc finger protein *ZFP36L1* (encoding the zinc finger protein) and *DEFA3* (encoding defensin) were discovered as new genes involved in bone metabolism. *RETN-CAP1* was discovered to participate in the interaction between immune cells and osteoclasts, showing that the osteoimmunology microenvironment significantly contributed to the pathogenesis of osteoporosis or osteopenia [54].

Due to the lack of human samples, further research should be undertaken in animal models to study osteoporosis. The primary purpose of murine model construction is to simulate the significant types of osteoporosis in humans, including post-menopausal osteoporosis, disuse osteoporosis, and glucocorticoid-induced osteoporosis [110]. As estrogen receptor α (Esr1) conditional knockout mice verified that estrogen promotes bone resorption and impairs osteoblast function [111,112,113,114,115,116,117], animal models are usually generated by ovariectomy, which eliminates estrogen secretion. Animal models for simulating disuse osteoporosis are generated by unloading the limb, specifically tail suspension or hind limb immobilization. In humans and large animals [118,119,120,121,122], but not always in rodents, glucocorticoid therapy significantly reduces cancellous bones, suppresses bone production, and promotes bone resorption. Future research on these animal models may fill the gap in the molecular mechanism of osteoporosis.

### 3.2. Cartilage- or Joint-Related Disorders

#### 3.2.1. Cartilage Injury

Cartilage has a limited intrinsic healing capacity due to the poor vascularization of tissue [123]. Identifying and characterizing regeneration-driving clusters and molecular mechanisms remain as research gaps. Previous research has shown that BMP9 induces a chondrogenic response in neonatal and adult amputations [124]. According to another pieces of research [125], fibroblasts are recognized as the most prevalent non-inflammatory mesenchymal cell types in amputation wounds. Together, these data suggest that a fibroblast is the sort of cell that responds to BMP9 and undergoes chondrogenesis during amputation regeneration. Using single-cell RNA datasets, Yu et al. confirmed the chondrogenic potential of a subgroup of amputation-wound-derived fibroblasts [70]. However, whether this result can be replicated in patients with simple cartilage injuries remains to be further explored.

#### 3.2.2. Osteoarthritis

Osteoarthritis (OA) has long been regarded as a degenerative illness that causes cartilage loss [126]. OA was formerly believed to be the result of any process that raised strain on a joint, such as loading on weight-bearing joints and anatomical joint incongruence, or fragility of the cartilage matrix caused by genetic modifications [126]. It took a decade for synovitis to be recognized as a crucial aspect of OA [126,127,128]. The application of scRNA-seq provided new discoveries in chondrocyte classification and provided connections between cartilage and synovitis.

Ji et al. [56] discovered seven molecularly characterized groups of chondrocytes in human OA cartilage, including four traditional populations named as proliferative chondrocytes (ProCs), fibrocartilage chondrocytes (FCs), prehypertrophic chondrocytes (preHTCs), and hypertrophic chondrocytes (HTCs), together with three new populations named as regulatory chondrocytes (RegCs), homeostatic chondrocytes (HomCs), and effector chondrocytes (ECs) [129,130,131]. Favorable genes were predominantly expressed in RegCs, HomCs, and ECs, but unfavorable genes were expressed in a high percentage of ProCs, FCs, and preHTCs. These findings indicate that the cell populations of the former may inhibit the development of OA, whereas the cell populations of the latter may promote OA development [56]. New biomarkers of cartilage progenitor cells (CPCs), a cell type that potentially differentiates into FC, were also identified [57]. Chou et al. [57] identified cell groupings in both synovium and articular cartilage, including twelve different cell types from synovium and seven from cartilage. Similar to Ji et al., cell types such as HomC, HTCs, preHTCs, RegCs, and FCs were identified. In addition, the researchers identified two unique subgroups of chondrocytes, reparative chondrocytes (RepCs) and prefibrochondrocytes (preFCs), according to their gene expression patterns. HomC or HTC chondrocyte cell types were enriched in intact cartilage, whereas FC, preFC, RegC, RepC, and preHTC chondrocyte cell types were enriched in injured cartilage. FCs were the major source of numerous OA-related proteases. HomCs and HTCs were the major sources of MMP3 and SERPINA1 in damaged cartilage, respectively. Wang et al. [71] aimed to compare OA with Kashin–Beck disease (KBD), a chronic, endemic osteochondropathy in which chondrocyte necrosis occurs in the growth plate and articular surface and results in growth retardation and secondary osteoarthritis [132]. The RegC population, markedly expanded in OA, could be identified by the known markers CHI3L1, AEBP1, PLIN2, and STEAP1. Two unique chondrocyte populations, HomCs and the novel MTCs marked by mitochondrial electron transport and the response to hydroperoxide, expanded in KBD samples relative to normal or OA samples [71]. Lv et al. identified cell clusters including the HomCs, RegCs, the stressed chondrocytes (StrCs), and the degenerative chondrocytes (DegCs). In injured cartilage, the cell proportion of HomCs decreased and that of StrCs and RegCs increased. Within StrCs, researchers found a chondrocyte cluster, namely the ferroptotic chondrocyte cluster, characterized by preferential expression of ferroptotic hallmarks and genes. Comprehensive GSVA recognized TRPV1 as an anti-ferroptosis biomarker in human OA cartilage, which was supported by further experiments revealing the murine OA model [72].

Apart from chondrogenic cells, Chou et al. [57] also detected synovium cell composition and articular cartilage cell–cell interactions and identified possible upstream growth factors and cytokines that might regulate the chondrocyte genes. Twelve cytokines, including TNF, IL6, IL1B, IL1A, etc., were exclusively expressed by synoviocytes but not chondrocytes, demonstrating that synovium is the origin of a large number of signals that regulate chondrocyte transcription in the progression of OA, instead of the cartilage.

Several single-cell experiments have described cell subsets in the OA synovial tissues, including synovial fibroblasts (SFs) and synovial immune cells [133,134,135]. In addition, mice models were applied to several studies on OA single-cell characterization [136,137,138,139,140]. The relationship between synovitis and OA is expected to be further studied with the help of single cell technology in the future.

#### 3.2.3. Rheumatoid Arthritis

Rheumatoid arthritis (RA) is a progressive and aggressive immunological condition that may result in decreased mobility and handicap. It is characterized by chronic synovitis, pannus development, joint degeneration, and nearby bone erosion. The pathogenic underpinning of rheumatoid arthritis is synovitis; thus, key target cells of RA are synovial fibroblasts (SFs) and synovial immune cells, and the activities of these cells result in the loss of articular cartilage and bone [141]. Recent advances in scRNA-seq technology have facilitated the characterization of the two distinct kinds of synovial cells mentioned above, especially in immune cells [142,143,144]. In this section, we mainly focus on the characterization of synovial fibroblasts in RA patients.

Stephenson et al. [28] distinguished the differences between CD55^+^ synovial fibroblasts and the CD90^+^ subpopulation. CD55^+^ fibroblasts localize to the intimal lining and are responsible for the production and turnover of synovial fluid. GO enrichment showed that CD55^+^ fibroblasts are involved in endothelial cell proliferation and reactive oxygen species’ responses. The majority of CD90^+^ fibroblasts are located in the lower synovial sublining layer, which is rich in modules associated with metallopeptidase activity and the formation of the extracellular matrix [28]. Zhang et al. [73] combined single-cell mass cytometry and transcriptomics, revealing *THY1(CD90)^+^ HLA-DRA*^hi^ sublining fibroblasts expanded in RA synovia. Immune cell dynamics were also profiled in this study [73].

Several cutting-edge techniques and animal models have also been used in the study of RA. As a proof-of-concept method for objective mRNA analyses at the site of inflammation in these chronic inflammatory illnesses, spatial transcriptomics has been applied to profile synovial samples from RA and spondyloarthritis (SpA) patients [20]. A scRNA-seq characterization of the mouse model of antigen-induced arthritis (AIA) was discussed by Gawel et al. [145].

#### 3.2.4. IVD Degeneration (IVDD)

The intervertebral disc (IVD), a joint with limited movement, is the segregation of vertebrae in the spine [146]. The disc consists of three parts: the nucleus pulposus (NP), which is located in the center of the disc and is highly hydrated, the annulus fibrosus (AF), which is elastic and fibrous, and the cartilaginous endplate (CEP), which connects the disc to the vertebral bodies. Intervertebral disc degeneration is a frequent cause of lower back pain, a major cause of disability, and a common finding in spine imaging that becomes more prevalent with age [146]. Currently, a number of single-cell studies have characterized the virtual clusters participating in the pathological process, as well as differences in different parts of the disc.

Zhang et al. [74] identified novel chondrocyte subsets in the nucleus pulposus. GSEA revealed upregulated ferroptosis signaling in two novel chondrocyte clusters, CPCs and HomCs, in the IVDD group compared to the controls. The ferroptosis pathway might participate in disc degeneration pathogenesis and serve as a new target for intervening IVDD [74]. Ling et al. [75] revealed the heterogeneity of NP cells in IVD, and thus observed that the inflammatory response NP cells and fibrocartilaginous NP cells make up a significant component of NP cells in the late stage of IVDD. The inflammatory response NP cells significantly expressed *ERG1* and *FGF1* and are shown to undergo inflammatory and endoplasmic reticulum (ER) responses, as well as fibrocartilaginous activity. Cell–cell interaction analysis between NP cells and immune cells also revealed that macrophages interact with NP cells by macrophage migration inhibitory factor and NF-kB signaling pathways [75]. Cherif et al. [76] analyzed differential gene expression among degenerative and non-degenerative AF and NP [76]. In addition, using healthy samples, Gan et al. [55] identified potential cell lineages in healthy IVD tissue that might promote the regeneration process, which could rescue degeneration. They identified four NP progenitor cell subsets, together with several chondrocytes or fibroblast clusters. Several clusters expressed protective or regenerative genes such as *EGF* and were involved in pathways such as the GAS signaling pathway [55]. These findings may have important implications for cell selection and design in tissue engineering or stem cell therapy for the treatment of disc degeneration [55].

### 3.3. Muscle- or Tendon-Related Disorders

#### 3.3.1. Tendon Injury

Like cartilage, tendons have a limited intrinsic healing capacity due to the poor vascularization of tissue, although might be affected by ectopic materials [147]. Research gaps remain in identifying and characterizing regeneration-driving clusters and molecular mechanisms. Using single-cell transcriptomics, Harvey et al. [81] identified a Tppp3^+^ cell population in which a fraction expresses Pdfgra. Injection of PDGF-AA protein stimulates the formation of new tenocytes, while Pdgfra inactivation contributes to the opposite effect, demonstrating that Tppp3^+^Pdgfra^+^ cells are tendon stem cells. Interestingly, Tppp3^−^ Pdgfra^+^ fibro-adipogenic progenitor (FAP) simultaneously exists in the tendon stem cell niche and generates fibrotic cells, indicating a potential genesis of fibrotic scarring in regenerating tendons. These findings explain why fibrosis arises in wounded tendons and highlight therapeutic difficulties for enhancing tendon regeneration without concurrently increasing fibrosis by ectopic PDGF. [81]

#### 3.3.2. Tendinopathy

Tendinopathy is a tendon degenerative disease accounting for over 30% of primary care consultations [15,148,149]. Resident tenocytes maintain the tendon matrix, and tendinopathy is connected with alterations in remodeling activity. [148] Developing therapies to address this difficulty requires an understanding of the crucial tenocytes, stem cells, and other cell subsets participating in tendinopathy.

Kendal et al. [15] used CITE-seq, which integrated surface proteomics with the gene expression modality of cells from tendinopathic and healthy human tendons. In response to chronic tendinopathy, tenocytes linked with microfibrils had higher expression of pro-inflammatory markers and PDPN (Podoplanin). Increased expression of IL-33, together with other chemokine and alarmin genes, was performed in diseased endothelium [15]. Garcia-Melchor et al. [82] profiled single-cell transcriptomes in tendinopathic tissues compared with healthy ones. They mainly focused on the interaction between T-cells and tenocytes, which increases T-cell activation through an auto-regulatory feedback loop, stimulates tenocytes to produce inflammatory cytokines and chemokines, changes collagen composition in favor of collagen 3, and modifies collagen composition [82]. This new understanding of tendinopathy may help us to develop new treatment approaches.

#### 3.3.3. Muscle Injury

Skeletal muscle injuries are prevalent disorders, particularly among athletes [150]. Mechanical injuries result from myofiber necrosis, hematomas, and inflammation and entail the disruption of connective tissues [151,152]. Inflammation and regeneration of muscles are contingent on the type, extent, and severity of the damage [151]. Muscle stem cells (MuSCs) are crucial for the lifelong maintenance of skeletal muscle homeostasis and regeneration [153]. The network among MuSC and immune cells, endothelial cells, and FAPs orchestrates the regeneration process [154]. Single-cell analysis recognized several muscle cell subtypes, especially in MuSCs [155], and this technique was applied to decode muscle injury and regeneration.

De Micheli et al. administered scRNA-seq and CyTOF to the muscles of adult mice at different time points after injury. They discovered that myogenic stem/progenitor cells exhibited heterogeneous expression of various syndecan proteins in cycling myogenic cells, indicating that syndecans may coordinate myogenic destiny control [78]. Dell’Orso et al. [79] compared scRNA-seq datasets from uninjured and regenerative MuSCs. After muscle injury, MuSCs were isolated by FACS 60 h later [79]. As a result, three unique clusters of MuSCs were identified in the injured muscle, with increased expression of metabolism-related genes [79]. Oprescu et al. [80] identified regeneration-related characteristics in three kinds of cells. Firstly, they found immune cells demonstrated early pro- and anti-inflammatory infiltration. Secondly, they found a MuSC cluster enriched for gene expression associated with immune cell complement activation, major histocompatibility class II antigens, and cathepsin family members. They also discovered the expression of myogenic genes in the muscle that was regenerating. In addition, pseudo-time trajectory suggests that activated, chemokine-expressing FAPs transform into Dpp4^+^ and Cxcl14^+^ cells in non-injured tissues [80].

#### 3.3.4. Muscular Dystrophy

Multinucleated muscle fibers, as well as MuSC, non-myogenic mesenchymal progenitors (such as FAPs), immune cells, endothelial cells, and other mononuclear cells make up the skeletal muscle, which is a complex heterogeneous tissue [156,157]. Dysfunction of these cell types leads to various diseases. Muscular dystrophies are hereditary, myogenic disorders characterized by gradual muscular atrophy and varying degrees of muscle weakness. Duchenne muscular dystrophy (DMD), the best-known form of muscular dystrophy, is a severe, progressive muscle-wasting illness. The disease is caused by mutations in DMD which abolish the production of dystrophin in muscle [158,159]. Although this disease has been well-characterized, single-cell methods may still add new knowledge in the future. Using single-cell level qPCR in mice hind limbs, Malecova et al. [77] evaluated subclusters in FAPs based on differential Tie2 and Vcam1 expression levels. Under further experimental measurements, the diaphragms of the duchenne muscular dystrophy (DMD) mouse model displayed a substantial decrease in the number of Tie2^high^ subFAPs and an increase in the fraction of Vcam1^+^ subFAPs. [77]

## 4. Treatments

### 4.1. Implants

Implants are frequently utilized in orthopedic surgery in order to provide support for bones or replace missing bones [160,161]. Materials implanted in the human body can be classified according to material types: metal materials, polymer materials, inorganic materials, composite materials, biomaterials, etc. Orthopedic implants possess certain important qualities, including biocompatibility, relevant mechanical properties, high corrosion and wear resistance, and osseointegration, which ensures implants’ safe and effective usage [162]. Bone formation along the implant surface is essential for the mechanical stability and lifespan of implants [162]. The cellular and molecular processes enabling peri-implant bone development are not fully comprehended, which calls for assessments from the single-cell analysis.

#### 4.1.1. Metal Implants

Titanium alloy is the most commonly used bone implant because of its biocompatibility. A mouse tibial implant model constructed by Vesprey et al. [163] is clinically comparable to joint replacement surgery for humans. After Pdgfra- and Ly6a/Sca1-expressing stromal cells (PαS cells) were identified, single-cell RNA-seq analysis revealed that PαS cells are quiescent in undamaged bone tissue but show markers of proliferation and osteogenic differentiation immediately after implantation surgery.

In order to reveal the immune microenvironment involved in biomaterial-mediated bone repair, Li et al. [164] used scRNA-seq data to profile cells around implant materials with various properties, including surface morphology, hydrophilicity, and cumulative release profile of ions. The osteoimmune microenvironment and cellular heterogeneity in controlling the bone regenerating process were clarified into four types of materials. Due to the large number of neutrophils and S100a8^hi^ macrophages that medical stainless steel (SS) implants recruit, osseointegration is eventually replaced by the formation of a fibrous capsule. Efficient integration was achieved with a mild immune response with a moderate number of mature, Ltf-, Pabpc1l-, and Capg-expressing neutrophils and highly differentiated macrophages around strontium-incorporated SLA (SR). In addition, the analyses of cell–cell interactions, which may increase bone formation by boosting the recruitment of BMSCs through the CXCL12/CXCR3 signal axis, elucidates the promotive function of neutrophils in the osseointegration of implants [164].

To summarize, in current studies, distinctions are described qualitatively for one or several materials. However, single-factor quantitative analysis for process parameters and microstructures, which is important for the design of implants, has not been characterized by single-cell analysis. For instance, according to previous studies, surface modifications enhance mechanical qualities, wear and corrosion resistance, and biocompatibility, and nanoscale surface design improves osseointegration and prevents infection [162]. The application of single-cell technology and even spatial omics allows for a more detailed analysis of the in vivo effects of materials with different parameters, the identification of new cell types, and the temporal and spatial features of local immune responses and the osseointegration process, as these features are not likely to be captured by bulk analysis.

#### 4.1.2. Polymer Implants

Polymer materials are usually designed to be biocompatible, having properties similar to those of human bone, such as a more appropriate modulus compared to hard metals, which helps the materials to better integrate into the human body [165]. Its radiolucent properties, i.e., transparency to X-rays, facilitate the analysis of X-ray and CT scan results [166]. Polymer materials also reduce the adverse reactions caused by metal implants, such as allergies, especially to heavy metals, as well as ion erosion caused by the release of harmful metal particles [167]. Further work is required to establish single-cell assessments on polymer implants as a substitution for bone or cartilage, whereas several articles have already reported the usage of polycaprolactone in muscle loss.

Cherry et al. [168] reported single-cell RNA-sequencing data collected from the site of implantation of either polycaprolactone (PCL) or an extracellular matrix (ECM)—a derived scaffold in a mouse model of volumetric muscle loss. In mast cells and dendritic cells among immune cells and in endothelial cells among non-immune cells, the PCL-treated group showed robustly increased TNF signaling. In contrast, endothelial cells from the ECM-treated group showed decreased TNF signaling. In addition, Myc signaling, a pathway associated with chronic wounds, decreased only in fibroblast precursor cells in the ECM-treated wound, indicating that reduced Myc signaling may have an impact on fibroblast differentiation and contribute to reduced fibrosis in biological scaffolds. The same group also characterized B-cell response to PCL, combining single-cell transcriptome and B-cell receptor sequencing [169,170]. B-cell receptor (BCR) sequencing is a high-throughput RNA-sequencing method targeting the highly variable antibody (derived from BCR) repertoire, allowing for the characterization of the adaptive immune response against implants.

Due to the capacity to deliver bioactive agents and control important material properties, polymeric implants are recommended in clinical scenarios involving osseointegration as well as bone growth and regeneration [171]. The spatial–temporal properties of implantation related cell types, including integrin ligands and growth-factor-expressing cells, need to be assessed using high throughput single-cell profiling. It is also crucial to apply single-cell studies to the characterization of larger animals such as dogs, sheep, and pigs. Apart from bone implants, some implants for cartilage or soft tissue repair, such as hydrogels [123,172] and tissue engineering materials, have the potential for single-cell research. It is also easier to separate and extract separate cells from soft tissue than with bone implants.

### 4.2. Stem Cell Therapies

As mentioned in the previous sections, the intrinsic healing capacity of cartilage and tendons is limited [123,147]. Since articular cartilage regeneration is improbable, the treatments for articular cartilage rely on external stimulation or transplantation. Reconstructive surgery with autologous tendons is commonly considered for severe tendon injuries; however, the post-surgical re-injury incidence is rather high, and the buildup of autologous tendons produces muscle weakness. These constraints have led to the quest for alternative solutions, which now include human embryonic stem cells (hESC) and induced pluripotent stem-cell-derived cells (iPSC) and tissues [173,174,175].

Wu et al. [176] identified certain WNTs and MITF genes as hub genes directing the off-target differentiation of hiPSC chondrogenesis into brain cells and melanocytes. Targeting WNTs and MITF removes these cell lineages, dramatically increasing the quantity and homogeneity of chondrocytes produced from hiPSCs [176]. The same group [173] described a variety of clinical procedures, including the one used to produce chondrocytes from human embryonic stem cells on a collagen I/III membrane. To compare these produced chondrocytes and human bone marrow stromal cells cultivated on membranes (hBMSC-M), scRNA-seq was performed on these cells with various states of human chondrogenic ontogeny or hBMSC-M. According to the experiments, the manufacturing process is predictable and creates mostly immature chondrogenic and chondroinductive cells [173]. These characterizations allow for the production of porcine articular cartilage with long-term repair potential.

Feng et al. [98] identified primitive and multipotent Lgr5^+^ interzone cells from embryonic mice joints, which have the potential for cartilage repair. To test this capacity, Lgr5^+^ interzone tissues were extracted from E13.5 embryos and were transplanted to the needle-punctured knees of 8-week-old mice. It turned out that the transplanted cells induced regeneration in the lesion. According to the research, Lgr5^+^ interzone cells have the ability to repair articular cartilage, which may open up new therapeutic possibilities [98].

Regarding tendons, Nakajima et al. [177] provided an iPSC-based technique for the development of a tendon injury treatment alternative. iPSC-tenocyte differentiation and torso differentiation share comparable developmental phases, including presomitic mesoderm (PSM), somites, sclerotome, and syndetome. To further characterize the developing cells and monitor the transcriptional dynamics during iPSC-syndetome differentiation, time-series samples were obtained on days 2, 4, 6, and 8, and single-cell RNA sequencing was performed on various derivatives of iPSC-syndrome differentiation. The findings indicated that iPSC-tenocytes were a homogenous population consisting of multiple cell cycle stages, whose transcriptional characteristics approximated those of native tenocytes, indicating that iPSC-tenocytes will serve as a treatment option for tendon damage [177].

### 4.3. Drugs

Osteoarthritis and osteoporosis, two types of musculoskeletal diseases that can be controlled by chemicals, drove the invention of many empirical drugs and targeted drugs [4,5,6]. Due to interpatient heterogeneity and disease complexity, potential medicines for osteoarthritis continue to fail in clinical trials despite being successful in animal models. Sahu et al. used a single-cell platform based on cytometry by time-of-flight (CyTOF) to outline the effects of BMS-345541 [178], an NF-κB pathway inhibitor, and kartogenin [11,179], a chondro-inductive small molecule. BMS-345541 elicited a uniform drug response in all patients, inhibiting the expression of p–NF-κB, HIF2A, and inducible NOS in various clusters of chondrocytes, as well as dramatically suppressing four senescent cell types such as *NOTCH1^+^STRO1^+^* chondroprogenitor cells. In contrast, only a few patients responded to kartogenin, which affected specific clusters of senescent cells, while its effect on chondroprogenitor cell populations has still been not clearly described [17].

The U.S. Food and Drug Administration has recently authorized a monoclonal antibody against sclerostin (encoded by the gene *SOST*) for the treatment of osteoporosis. However, Ayturk et al. [62] failed to discover a discernible difference in the relative cell subcluster proportions between the long bone of sclerostin-neutralizing antibody-treated and control mice. The most plausible explanation is that the cell recovery protocol was insufficiently sensitive to detect moderate gene expression changes. In order to develop scRNA-seq techniques that are highly sensitive to identify subtle changes in gene expression in response to genetic, pharmacological, or environmental perturbations, more effective osteoblast enrichment protocols will be required in the future [62].

There is abundant room for further application of single-cell techniques in drug assessment in other diseases. Manufacture of targeted medicine is bound to be the development direction of disease treatment. Single-cell technologies enable precise analysis of the efficacy of traditional drugs in different individuals. It may also lead to a new understanding of the therapeutic mechanisms caused by these drugs. The single-cell and spatial omics analyses of the therapeutic mechanisms provide the research and development of targeted drugs with another perspective.

## 5. Emerging Directions for Single-Cell Profiling of Musculoskeletal Diseases

As mentioned above, single-cell sequencing and spatial omics technologies have played an essential role in the characterization of gene expression and cellular compositions in many musculoskeletal diseases, including skeletal-related, cartilage-related, and muscle-related diseases. However, studies have only been carried out in a limited number of musculoskeletal disorders, and experimental design requires further improvements according to the characteristics of the diseases. For example, studies on defects in cartilage and tendon development still need to be included; single-cell analysis on ectopic ossification has yet to be extended to human clinical samples; studies on osteoporosis are absent in clinical disease samples and in murine models. More single-cell and spatial omics characterization would help us establish a greater degree of accuracy on these topics. 

Since distinct cell types enable the musculoskeletal system to produce a broad variety of responses in health and disease, improved delineation of cellular populations is crucial for single-cell investigations. Making such discoveries using single-cell transcriptomics is not only common but also feasible. Advanced developments in single-cell technologies, which enable the combined investigation of muscular skeletal lineage repertoires and molecular states, are expected to improve our knowledge of the musculoskeletal system and to reveal the way the remodeling processes take place during disease [180]. Several methods for profiling single-cell epigenomic features (single-cell assay for transposase-accessible chromatin with high-throughput sequencing, scATAC-seq [181,182]) and proteomics (cytometry by time-of-flight, CyTOF [18]) contribute to different sets of profiles in the musculoskeletal system. Recent research has shown that multi-omics has advanced from the single-modal assessment of transcriptomes to massively parallel single-cell omic measurements [183], simultaneously profiling two or all three modalities among transcriptomes, proteomes, and epigenomes [16,184,185,186,187,188,189]. We will be able to define more accurate cellular phenotypes and examine their roles in the musculoskeletal system in both health and disease thanks to these integrated strategies. Furthermore, spatial omics has advanced to spatial epigenomics, which presents fresh prospects for studying epigenetic regulation, cell function, and destiny determination in healthy physiology and disease [190].

Moreover, therapy-related single-cell research also has a huge scope for exploration. With the wide application of bioactive materials to cure musculoskeletal disorders, many researchers have focused on the biological function of materials themselves. Single-cell technology can help to promote the researchers’ new understanding of existing materials and the selection of new materials. It is also expected to provide ideas for the design of tissue engineering materials. In the field of stem cell therapy, single-cell technology has the ability to directly explain its role, which is significant in promoting the approval of stem cell therapies by the FDA. In addition, in the development of new drugs for the musculoskeletal system, single-cell technology can help developers better judge the effects of drugs during the research phase and avoid unnecessary clinical trials. Overall, single-cell and spatial omics techniques have important and far-reaching implications for improving the therapeutics of the musculoskeletal system by enhancing the understanding of the effective mechanisms of therapeutic approaches.

## Figures and Tables

**Figure 1 ijms-24-02271-f001:**
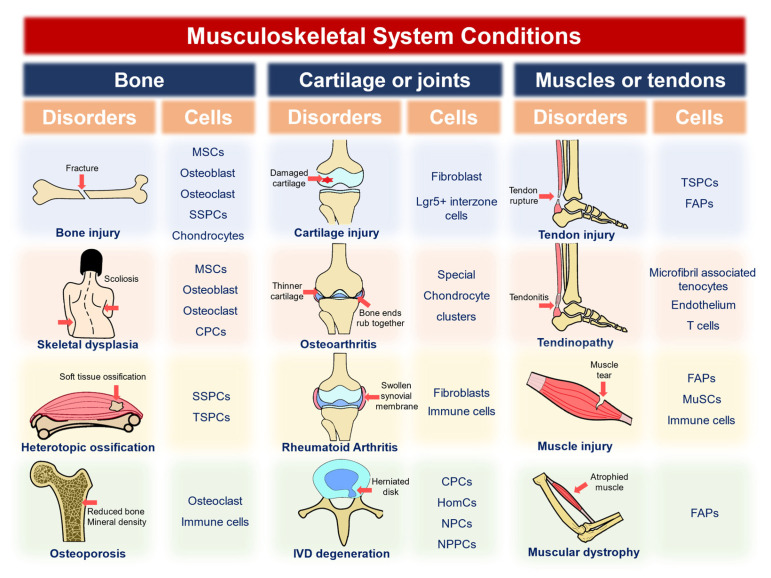
Musculoskeletal system conditions and the cell populations involved have been studied by single-cell or spatial omics techniques. MSCs: mesenchymal stem cells; SSPCs: skeleton stem/progenitor cells; CPCs: cartilage progenitor cells; NPCs: nucleus pulposus cells; NPPCs: NP progenitor cells; TSPCs: tendon stem/progenitor cells; HomCs: homeostatic chondrocytes; FAPs: fibro-adipogenic progenitors; MuSCs: muscle stem cells.

**Table 1 ijms-24-02271-t001:** Common cell types and markers of the musculoskeletal system.

Cell Type	Transcript Markers	Reference
MSC	*ITGB1, VACAM1, THY-1, NT5E, ENG*	Yang et al., 2021 [52]
Embryonic SSC	*PDPN, CADM1*	He et al., 2021 [53]
Cranial neural crest cells	*PDPN, CADM1*	He et al., 2021 [53]
Osteoblast	*RUNX2, COLA1, SPP1, ANO5, CDH11*	Yang et al., 2021 [52], Wang et al., 2022 [54]
Chondrocyte	*COL2A1, SOX9, COL10A1, Aggrecan, COMP,*	Yang et al., 2021 [52], Gan et al., 2021 [55]
CPC	*BIRC5, UBE2C, DHFR, CENPU, STMN1*	Yang et al., 2021 [52], Ji et al., 2019 [56]
Prehypertrophic/hypertrophic chondrocytes	*COL10A1, IBSP*	Chou et al., 2022 [57]
Osteoclast	*NFKB-1, NFKB-2, ATP6V0D1, NFATC1, OSCAR, MMP9, MMP8*	Yang et al., 2021 [52], Wang et al., 2022 [54]
Tendocyte	*COL1A1/2*	Kendal et al., 2020 [15]
MuSC	*APOE, PAX7, MYF5, APOC*	De Micheli et al., 2020 [58]
Mature skeletal muscle cells	*TTN, MYLPF, CKM, TNNC, ACTA1*	De Micheli et al., 2020 [58]
Adipocytes	*APOD, CXCL14*	De Micheli et al., 2020 [58]
Periosteal Osteogenic Progenitors	*LEPR, PRRX1, GREM1*	Ding et al. 2022 [59]

MSC: mesenchymal stem cell; CPC: chondrocyte progenitor cell; SSC: skeletal stem cells; MuSCs: muscle stem cells.

**Table 2 ijms-24-02271-t002:** Application of single-cell and spatial omics to decode bone-related diseases.

Disease	Animal	Location	Phenotype Driving Gene/Materials in Mice Models	Important Affected Cell Types	Reference
Bone injury and regeneration	Mice	Rib	Smo	Cxcl12-expressing SSPCs	Serowoky et al., 2022 [64]
	Mice	Frontal bones	p75 (*Ngfr*)	Itgb1-expressing mesenchymal, Il1a-, Il10-, and Tnf-expressing mesenchymal, and immune cells	Xu et al., 2022 [65]
	Mice	Long bone	-	Osteoblast lineage cells, chondrocytes, fibroblasts, Fabp5^+^ Mmp9^+^ septoclasts	Sivaraj et al., 2022 [66]
Congenital Skeletal dysplasia	Mice	Hind limb	-	HC	Wang et al., 2022 [67]
	Mice	Calvarium	TrkA	Mesenchymal progenitor cells	Tower et al., 2021 [19]
Adolescent idiopathic scoliosis	Human	Spinal cancellous bone tissues	-	MSC-IGFBP5, CPC-PCNA, and OC-BIRC3	Yang et al., 2021 [52]
Heterotopic ossification	Mice	Muscle	rhBMP2–Matrigel mixtures pre-immune antibody (referred to as BMP2/IgG) or neutralizing activin A antibody (BMP2/nActA.Ab)	Sox9-expressing skeletal progenitors, and Acan- and Col2a1-expressing clusters	Mundy et al., 2021 [68]
	Mice	Achilles tendon	-	Prg4+ TSPC	Tachibana et al., 2022 [69]
Osteoporosis	Human	Femoral head	-	Osteoclasts and immune cells	Wang et al., 2022 [54]

SSPCs: skeleton stem/progenitor cells, HC: hypertrophic chondrocytes, MSC: mesenchymal stem cell, CPC: chondrocyte progenitor cell, OC: osteoclast, TSPCs: tendon stem/progenitor cells.

**Table 3 ijms-24-02271-t003:** Application of single-cell and spatial omics to decode cartilage or joint-related diseases.

Disease	Animal	Location	Phenotype Driving Gene/Materials in Mice Models	Important Affected Cell Types	Reference
Chondrogenic regeneration	Mice	Digit tip	Ectopic BMP9	Fibroblasts	Yu et al., 2022 [70]
Osteoarthritis	Human	Cartilage	-	ProC, FC, preHTC, and CPC	Ji et al., 2019 [56]
	Human	Cartilage	-	FC, preFC, RegC, RepC, and preHTC	Chou et al., 2020 [57]
	Human	Cartilage	-	RegC	Wang et al., 2021 [71]
	Human	Cartilage	-	StrC (containing a ferropotic cluster), RegC	Lv et al., 2022 [72]
	Human	Synovitis	-	Immune cells and fibroblasts	Chou et al., 2020 [57]
Kashin–Beck disease	Human	Cartilage	-	HomC and MTC	Wang et al., 2021 [71]
Rheumatoid arthritis	Human	Synovitis	-	CD55^+^ fibroblast and CD90^+^ fibroblast	Stephenson et al., 2018 [28]
	Human	Synovitis	-	THY1(CD90)+HLA-DRAhi sublining fibroblasts	Zhang et al., 2019 [73]
	Human	Synovitis	-	Immune cells fibroblasts	Carlberg et al., 2019 [20]
IVD degeneration	Human	IVD	-	CPC, HomC, and other chondrocyte subsets	Zhang et al., 2021 [74]
	Human	IVD	-	IR NPC, FC NPC	Ling et al., 2021 [75]
	Human	IVD	-	-	Cherif et al.,2022 [76]
	Human	IVD	-	NP progenitor cells	Gan et al., 2021 [55]

proC: proliferative chondrocytes, FC: fibrochondrocytes, preHTC: prehypertrophic chondrocytes. CPC: cartilage progenitor cells, preFC: prefibrochondrocytes, RegC: regulatory chondrocytes, RepC: reparative chondrocytes, StrC: stressed chondrocytes, HomC: homeostatic chondrocytes, MTC: mitochondrial chondrocytes, NP: nucleus pulposus, IR NPC: inflammatory response NP cells, FC NPC: fibrocartilaginous NP cells.

**Table 4 ijms-24-02271-t004:** Application of single-cell and spatial omics in muscle- or tendon-related diseases.

Disease	Animal	Location	Phenotype Driving Gene/Materials in Mice Models	Important Affected Cell Types	Reference
Muscular dystrophy	Mice	Single-cell analysis: hind limb; further experiment: diaphragms	Mdx	Tie2^high^ FAP and Vcam^+^ FAP	Malecova et al., 2018 [77]
Injury of muscle	Mice	Hind limb muscle	Notexin	MuSC, myogenic progenitors, FAP, and tendocytes	De Micheli et al., 2020 [78]
	Mice	Hind limb muscle	Notexin	MuSC, PM	Dell’Orso et al., 2019 [79]
	Mice	Tibialis anterior (TA) muscles	Notexin	MuSC-expressing immune genes, immune cells, and activated FAP	Oprescu et al., 2020 [80]
	Mice	Patellar tendons	PDGF-AA protein	Tppp3^+^Pdgfra^+^ tendon stem cells. Tppp3^−^Pdgfra^+^ FAP	Harvey et al., 2019 [81]
Injury of tendon	Mice	Diseased chilles, toe extensor or diseased peroneus longus	-	IL33-expressing endothelium and microfibril-associated tenocytes	Kendal et al., 2020 [15]
Tendinopathy	Human	Supraspinatus and subscapularis tendon	EGF and GAS pathway	T-cells	Garcia-Melchor et al., 2021 [82]

FAP: fibro-adipogenic progenitor, MuSC: muscle stem cells, PM: primary myoblasts, PDGF-AA: platelet-derived growth factor-AA.

## Data Availability

Not applicable.

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
