# Peer review of "Application of Single-Cell and Spatial Omics in Musculoskeletal Disorder Research"

_ijms, 2023, doi:10.3390/ijms24032271_

Round 1

Reviewer 1 Report

Feng et al reviewed the current and prospective landscape of single cell and spatial omics to provide a comprehensive picture in musculoskeletal disorders. The manuscript is good for the journal. Before the manuscript can be published, the authors should address the following issues.

  1. The musculoskeletal system is composed of a complex mixture of many dynamic cell types. Table 1, the authors introduced a few key cell types, however it should be expanded, such as the muscle tissue (DOI: 10.1186/s13395-020-00236-3), adipo tissue, periosteum tissue, prehypertrophic/hypertrophic chondrocytes, SSCs, cranial neural crest cells.

  2. In section 2, the authors can consider adding the analysis approaches, tools, and their limitations of single-cell and spatial omics that will make the review more comprehensive.

  3. In terms of spatial omics, for example, the GeoMx DSP technology from nanoString, MERFISH, and STARmap are currently widely adopted for disease concepts. The authors can expand the discussion of these spatial biology technologies.

Reviewer 2 Report

In this review article, the authors tried to summarize the utility, present knowledge and future perspectives of single-cell and spatial omics in the fields of bone, cartilage, muscle and tendon research and relevant clinical sciences.  This is a unique attempt to overview a number of studies belonging to a wide range of medical and biological category and thus precious.  However, several points need to be addressed.

Major points

1) The title does not correctly represent the contents of the article.  Application of something to disorders suggests therapeutic utility of something for disorders.  For example, application of single-cell and spatial omics in musculoskeletal disorder research may be better.  Relevant phrases in Abstract should be reconsidered as well.  

2) Figure 1 needs reconsideration.  First, a number of abbreviations in the legend, such as FCs, preFCs, RegCs, RepCs, preHTCs and so on, do not appear in the figure body.  Second, the three categories shown in Prussian blue should be represented by the words in the same category: for example, Bone, Cartilage or joint, Muscle or tendon.  Third, all of the words below "Disorders" and "Cells" should be enlarged. 

3) The table also contains multiple problems.  Although this is the only table in this article, it is labeled Table 2.  Involvement of too many columns and lines makes this table hard to follow.  This table needs to be split into 3 pieces; the first one for "osteogenic (or bone)", the second one for "chondrogenic or joint (or cartilage or joint)", and the third for "Muscle and tendon"; then the first column can be removed.  Columns for "Resources of samples" and "Methods" may be omitted, since details are described in the text.  As a result, three concise tables that are easy to follow can be obtained.  

4) Several abbreviations are used undefined or defined after the first use in the text.  What does "CITE" stand for?  As a latter example, IVD is defined on page 14. 

5) The last paragraph of section 4 and the first paragraph of section 5 are exactly the same.  One of them should be removed and the corresponding section should be rewritten.  

Minor points

3.2.4. IVD degradation (IVDD), fourth sentence from the last: Do the authors mean "cell lineages" rather than "cell lines"?

3.1.2. Skeletal dysplasia: This subsection may not necessarily be divided into further subsections.

4.1.2. Polymer Implants, second paragraph, the last sentence: What is "B cell receptor sequencing"?  Please explain it. 

4.2. Drugs, fifth paragraph, second sentence: What this sentence "However, single-cells studies still need to be improved in the study of quite a few diseases of the skeletal-muscular system." is not clear.  Please rephrase it.  

References: The ones after No. 61 are not exactly following the style of this journal.

Round 2

Reviewer 2 Report

Most of the issues pointed out have been addressed; however, several problems are still left, or have newly emerged.  

Specific points

1) Usually, "-related" does not follow the plural form of a noun.  It is recommended to use "muscle or tendon-related" and "cartilage or joint-related" instead of "muscles-or-tendons-related" and "cartilage-or-joints-related". 

2) Figure 1 has been improved, but abbreviations NPCs and NPPCs are not defined in the legend.  

3) Tables 2, 3 and 4 have been improved.  However, large tables split in two pages are not reader friendly.  It is recommended to reduce the font size used in the tables up to that used for the title of the tables.

4) What are "Itgb1 mesenchymal, Il1a, Il10 and Tnf-expressing mesenchymal, immune cells" in Table 2?  Do the authors mean "Itgb1 mesenchymal, Il1a, Il10 and Tnf-expressing mesenchymal, and immune cells"?

5) Unfortunately, all of the references have been unified in a wrong format.  Please check carefully the format given in the IJMS template and modify them properly.

Author Response

Most of the issues pointed out have been addressed; however, several problems are still left, or have newly emerged. 

Specific points

1) Usually, "-related" does not follow the plural form of a noun.  It is recommended to use "muscle or tendon-related" and "cartilage or joint-related" instead of "muscles-or-tendons-related" and "cartilage-or-joints-related".

Response:

Thanks for your suggestion.  We have used "muscle or tendon-related" and "cartilage or joint-related" instead of "muscles-or-tendons-related" and "cartilage-or-joints-related". You can see the changes in the subheadings and table names

2) Figure 1 has been improved, but abbreviations NPCs and NPPCs are not defined in the legend. 

Response:

Thanks for your suggestion. We have defined NPCs and NPPCs in the legend.

You can see that in line 260:

“NPCs: nucleus pulposus cells; NPPCs: NP progenitor cells”

3) Tables 2, 3 and 4 have been improved.  However, large tables split in two pages are not reader friendly.  It is recommended to reduce the font size used in the tables up to that used for the title of the tables.

Response:

Thank you for your advice. We changed the font of the tables so that each table is on the same page to ensure that it is easy to read.

4) What are "Itgb1 mesenchymal, Il1a, Il10 and Tnf-expressing mesenchymal, immune cells" in Table 2?  Do the authors mean "Itgb1 mesenchymal, Il1a, Il10 and Tnf-expressing mesenchymal, and immune cells"?

Response:

Thank you for your comment. We corrected the description to " Itgb1-expressing mesenchymal, Il1a, Il10 and Tnf-expressing mesenchymal, and immune cells ".

5) Unfortunately, all of the references have been unified in a wrong format.  Please check carefully the format given in the IJMS template and modify them properly.

Response:

Thanks for your suggestion, we have checked and revised the reference format again with a bibliography software package called Sciwheel.

e.g.

  1. Gilliland, K.O.; Kernick, E.T. Musculoskeletal tissues and anatomy. In Clinical foundations of musculoskeletal medicine: A manual for medical students; Esther, R. J., Ed.; Springer International Publishing: Cham, 2021; pp. 11–21 ISBN 978-3-030-42893-8.
  2. Salhotra, A.; Shah, H.N.; Levi, B.; Longaker, M.T. Mechanisms of bone development and repair. Rev. Mol. Cell Biol. 2020, 21, 696–711, doi:10.1038/s41580-020-00279-w.
  3. Heath, J.R.; Ribas, A.; Mischel, P.S. Single-cell analysis tools for drug discovery and development. Rev. Drug Discov. 2016, 15, 204–216, doi:10.1038/nrd.2015.16.